# Simultaneous quantification of palbociclib, ribociclib and letrozole in human plasma by a new LC-MS/MS method for clinical application

**Bianca Posocco**[1]*, **Mauro Buzzo**[1], **Ariana Soledad Poetto**[1,2], **Marco Orleni**[1],
**Sara Gagno**[1], **Martina Zanchetta**[1,3], **Valentina Iacuzzi**[1,4], **Michela Guardascione**[1],
**Fabio Puglisi**[5,6], **Debora Basile**[5,6], **Giacomo Pelizzari**[5,6], **Elena Marangon**[1],
**Giuseppe Toffoli**[1]

1 Experimental and Clinical Pharmacology Unit, Centro di Riferimento Oncologico di Aviano (CRO) IRCCS, Aviano, Italy, 2 Doctoral School in Pharmacological Sciences, University of Padova, Padova, Italy, 3 Department of Chemical and Pharmaceutical Sciences, University of Trieste, Trieste, Italy, 4 Doctoral School in Nanotechnology, University of Trieste, Trieste, Italy, 5 Department of Medicine (DAME), University of Udine, Udine, Italy, 6 Department of Medical Oncology, Centro di Riferimento Oncologico di Aviano (CRO) IRCCS, Aviano, Italy

* bposocco@cro.it

**Data Availability Statement:** All relevant data are within the manuscript and its Supporting Information files.

## Abstract

A novel LC-MS/MS method was developed for the quantification of the new cyclin dependent kinase inhibitors (CDKIs) palbociclib and ribociclib and the aromatase inhibitor letrozole used in combinatory regimen. The proposed method is appropriate to be applied in clinical practice due to the simple and fast sample preparation based on protein precipitation, the low amount of patient sample necessary for the analysis (10 μL) and the total run time of 6.5 min. It was fully validated according to FDA and EMA guidelines on bioanalytical method validation. The linearity was assessed ($R^2$ within 0.9992–0.9983) over the concentration ranges of 0.3–250 ng/mL for palbociclib, 10–10000 ng/mL for ribociclib and 0.5–500 ng/mL for letrozole that properly cover the therapeutic plasma concentrations. A specific strategy was implemented to reduce the carryover phenomenon, formerly known for these CDKIs. This method was applied to quantify the $C_{min}$ of palbociclib, ribociclib and letrozole in plasma samples from patients enrolled in a clinical study. The same set of study samples was analysed twice in separate runs to assess the reproducibility of the method by means of the incurred samples reanalysis. The results corroborated the reliability of the analyte concentrations obtained with the bioanalytical method, already proved by the validation process. The percentage differences were always within ±10% for all the analytes and the $R^2$ of the correlation graph between the two quantifications was equal to 0.9994.

## Introduction

Palbociclib (PALBO) and ribociclib (RIBO) are orally bioavailable, small molecule cyclin dependent kinase inhibitors (CDKIs) indicated for the treatment of hormone receptor (HR)-positive, human epidermal growth factor receptor 2 (HER2)-negative locally advanced or

**Funding:** The authors received no specific funding for this work.

**Competing interests:** I have read the journal's policy and the authors of this manuscript have the following competing interests: Dr. Fabio Puglisi reports grants from Astrazeneca and from Roche, outside the submitted work. This does not alter our adherence to PLOS ONE policies on sharing data and materials.

metastatic breast cancer in combination with an aromatase inhibitor such as letrozole (LETRO) [1,2]or with the oestrogen receptor antagonist fulvestrant in patients who have received prior endocrine therapy [3].

According to the recommended dosing regimen PALBO and RIBO are administered orally once a day at a dose of 125 mg (one capsule) and 600 mg (three tablets) respectively. The therapy is taken for 21 consecutive days followed by 7 days off treatment (28-day cycle) in combination with LETRO 2.5 mg/day given continuously (or fulvestrant 500 mg by intramuscular injection on days 1, 15, and 29 of cycle 1 and once a month for the following cycles, in the case of PALBO).

Variability in response and/or toxicity towards anticancer drugs could be related to variability in pharmacokinetic parameters such as area under the plasma concentration *vs* time curve (AUC) or plasma trough level ($C_{min}$) [4,5]. Therefore, pharmacokinetics could be an accessible biomarker for therapy optimization through therapeutic drug monitoring (TDM) identifying patients at risk of toxicity due to high exposure or who may experience suboptimal efficacy due to low exposure. Moreover, TDM allows to highlight cases of suspected non-adherence to therapy, which is a particularly plausible issue as far as oral drugs are concerned.

Based on the limited exposure–response and–toxicity studies, TDM recommendation for PALBO, RIBO and LETRO, currently ranges from exploratory to promising. A greater reduction in absolute neutrophil count appears to be associated with increased PALBO exposure, while no definitive exposure-response relationship was found in 81 patients treated at 125 mg fixed dose [6]. Some adverse events, as neutropenia or QT prolongation, have shown to be proportional to RIBO exposure [7]. Finally, an exposure-efficacy analysis in patients treated with LETRO showed a longer time to tumour progression for those patients achieving LETRO plasma concentrations $\geq$ 85.6 ng/mL [5].

To deepen the knowledge about the interindividual variation in pharmacokinetics and relationship with patient outcome, large prospective clinical studies are needed, as well as more validated bioanalytical methods to support them. At the best of our knowledge, for the quantification of these drugs in human plasma, only one LC-UV method was published for PALBO [8], two LC-MS/MS methods were published for RIBO [9,10], while several LC-MS/MS assays [11–16] and one LC-UV method [17] were reported for LETRO. Recently, a LC-MS/MS method was also proposed for the quantification of the CDKIs in human plasma [18] but not for the simultaneous determination of LETRO. Moreover, the calibration range was 2–200 ng/mL for all the analytes and, thus, it does not properly cover all the *in vivo* concentration range, especially for RIBO. As reported in the literature, *in vivo* RIBO plasma concentrations (CV) fall in between 711 (72.9) ng/mL ($C_{min}$) and 3500 (65.8) ng/mL [19,20] while PALBO plasma concentrations range from the population mean $C_{min}$ (CV) of 61 (42) ng/mL [6] to 185.5 (27) ng/mL ($C_{max}$, day 8 of cycle 1 at standard dose) [21].

With regard to LETRO, Beer et al. reported a mean concentration (±SD) of 107.0±62.9 ng/mL [11] while from the study of Desta et al. a mean concentration (range) of 89.7 (28.4–349.2) ng/mL was obtained [22].

The aim of this work was to develop and validate a new LC-MS/MS method to simultaneously monitor plasma concentrations of PALBO, RIBO and LETRO in human plasma. The proposed assay was used to perform $C_{min}$ quantifications in breast cancer patients.

## Materials and methods

### Chemicals and reagents

The analytical standard of PALBO was provided by Toronto Research Chemicals (Toronto, Canada, product N. P139900, lot N. 1-ZPK-125-1), RIBO hydrochloride (product N. M15373, lot

N. 12195) and LETRO (product N. L6545, lot N. 0000028530) were supplied by Merck-Sigma Aldrich. Stable isotopically labeled internal standards $D_6$-RIBO (product N. C4503, lot N. PO-ALS-18-007-B1), $D_8$-PALBO (product N. C5108, lot N. SA-ALS-15-120-P1) and $^{13}C_2,^{15}N_2$-LETRO (product N. C595, lot N. JA-ALS-18-104-P3) were synthesized by Alsachim (Illkirch Graffenstaden, France). LC-MS grade isopropanol was supplied by Merck-Sigma Aldrich while LC-MS grade methanol was purchased from Carlo-Erba (Milano, Italy). "Type 1" ultrapure water was produced at our department by a Milli-Q® IQ 7000 system (Merck). Drug-free plasma/K-EDTA from healthy volunteers to prepare daily standard calibration curves and quality control samples (QCs) was provided by the transfusion unit of our institution.

## Standard solutions preparation

Stock solutions of RIBO and LETRO were prepared in methanol at the concentration of 1 mg/mL while stock solution of PALBO was prepared in DMSO at 0.5 mg/mL. Two different stock solutions were obtained for each compound: one for the calibration curve and the other for QCs. To obtain the working solutions for the construction of the calibration curve (from A to H) the stock solutions of PALBO, RIBO and LETRO were mixed together and diluted with methanol to achieve the final concentrations of: 5, 3, 1.5, 0.5, 0.2, 0.08, 0.02, and 0.005 μg/mL for PALBO, 200, 120, 60, 20, 8, 3.2, 0.8, and 0.2 μg/mL for RIBO, and 10, 6, 3, 1, 0.4, 0.16, 0.4, 0.16, 0.04, and 0.01 μg/mL for LETRO. The stock solutions for QCs (H-high, M-medium, L-low) were mixed together and diluted with methanol to obtain the final concentrations of: 4, 0.4, 0.01 μg/mL for PALBO, 160, 16, 0.4 μg/mL for RIBO, and 8, 0.8, 0.02 μg/mL for LETRO. Stock solutions of IS were prepared in methanol for $D_6$-RIBO and $^{13}C_2,^{15}N_2$-LETRO at the concentrations of 1 and 0.5 mg/mL, respectively, and for $D_8$-PALBO in DMSO at 0.5 mg/mL. The three working solutions were mixed together and diluted with methanol to obtain the final concentrations of 12.5 ng/mL for $D_8$-PALBO and $^{15}N_2$-LETRO, and 45.0 ng/mL for $D_6$-RIBO. This solution was directly used to precipitate plasma proteins during sample treatment.

## Calibration curve, QCs and patients' sample preparation

Preparation of calibration curve and QCs samples was conducted through the following steps: 1) 5 μL of working solutions were added to 95 μL of blank human plasma (dilution 1:20) and vortexed for 10 s; 2) a 10 μL-aliquot of this mix was added with 80 μL of cold IS working solution (see "Standard solutions preparation" section), vortexed and then centrifuged for 15 min at 16200 g and 4°C; 3) 70 μL of the supernatant were transferred to a polypropylene tube for the following analysis. The final concentrations thus obtained were: 0.3, 1, 4, 10, 25, 75, 150, 250 ng/mL for PALBO, 10, 40, 160, 400, 1000, 3000, 6000, 10000 ng/mL for RIBO, 0.5, 2, 8, 20, 50, 150, 300, 500 ng/mL for LETRO for the calibration curve; 0.5, 20, 200 ng/mL for PALBO, 20, 800, 8000 ng/mL for RIBO, and 1, 40, 400 ng/mL for LETRO for the QCs. The preparation of patients' samples was conducted as following: patient plasma was thawed at room temperature, vortexed for 10 s and centrifuged, for 10 min, at 3000 g and 4°C; 10 μL of plasma was then treated according to steps 2) and 3).

## Chromatographic and mass spectrometry conditions

The HPLC system was a SIL-20AC XR auto-sampler and LC-20AD UFLC XR pumps (Shimadzu, Tokyo, Japan). The mobile phases (MP) consisted of ultrapure water with 0.1% HCOOH (phase A) and methanol/isopropanol (9:1, v/v) with 0.1% HCOOH (phase B). The chromatographic separation was obtained on the Luna Omega Polar C18 column (3 μM, 100 Å, 50 x 2.1 mm) coupled with a Security Guard Cartridge (Polar, C18, 4 x 2.0 mm), both provided by Phenomenex (Castel Maggiore (BO), Italy). The mass spectrometry system used for

the detection was an API 4000 triple quadrupole (AB SCIEX, Massachusetts, USA) with a TurboIonSpray source operating in positive ion mode. To optimize source and compound dependent parameters, solutions of each analyte at the concentration of 100 ng/mL were used with a flow rate of 20 μL/min. Data were processed with Analyst 1.6.3 and the quantification of the peaks was done with MultiQuant 2.1 (software package AB SCIEX).

## Method validation

A full validation of the proposed method was conducted according to FDA and EMA guidelines on bioanalytical method validation [23,24], as previously reported [25,26].

**Recovery, matrix effect and selectivity.** Three different sets (set 1, 2, 3) of QCs were prepared in quintuplicate at each concentrations (L, M, H): set 1) normal QCs prepared as reported in "Calibration curve, QCs and patients' sample preparation" section; set 2) post-extraction QCs (QC working solution was added to an extracted plasma sample); set 3) QCs in pure methanol. To evaluate PALBO, RIBO and LETRO recovery, the peak area ratio of set 1 over set 2 QCs was calculated.

Effects of matrix endogenous components on the ionization of PALBO, RIBO and LETRO were evaluated with different strategies during the chromatographic method development and, successively, during the validation process. Firstly, this phenomenon was investigated by means of the post-column infusion using standard solutions of the three analytes in 0.1% HCOOH methanol/water 1:1 at the concentration of 50 ng/mL and applying a flow rate of 20 μL/min. For more details on post-column infusion experiment see our previous published methods [25,26]. The matrix effect was then evaluated by calculating for each analyte the ratio of the peak area of set 2 QCs to the peak area of set 3 QCs. The CV should be within 15% [24].

To investigate the selectivity of the proposed bioanalytical method (i.e. the presence of interferences whose signal overlaps with those of the analytes of interest), 6 blank human plasma samples obtained from 6 different healthy donors were analysed. The samples analyzed should be free of interference at the retention times of the analytes of interest. The absence of interference was defined as a response lower than 20% of the LLOQ for the analytes and lower than 5% for the ISs.

**Linearity and sensitivity.** Calibration curves were built using a weighted ($1/x^2$) linear regression model. To evaluate the linearity of the curve, 5 calibration curves freshly processed during different working days were used. The Pearson's determination coefficient $R^2$ was calculated and the comparison of the true and back-calculated calibration standard concentrations (expressed as accuracy) was checked. A minimum of 7 out of 8 calibration points, including the lower limit of quantification (LLOQ) and the highest calibrator (ULOQ), had to be within 85–115% of the theoretical concentration (80–120% at the LLOQ) [23,24].

The sensitivity of a bioanalytical method is defined by the LLOQ. The LLOQ is the lowest concentration that could be measured with a precision (i.e. the coefficient of variation (CV%), expressing the standard deviation as a percentage of the mean calculated concentration) within 20%, accuracy between 80% and 120% and a signal-to-noise ratio (S/N) $\geq$ 5. The LLOQ of the present method was verified analysing the precision, accuracy and S/N ratio obtained from 6 samples of pooled blank human plasma added with H working solution (prepared as reported in "Calibration curve, QCs and patients' sample preparation" section). The Analyst software calculates the S/N ratio using Peak-to-Peak method taking the standard deviation of all the chromatographic data points between the specified background start and background end times (60 min before analyte peak).

**Carryover.** Since previously published methods [18,27] underlined the presence of carryover effect related to both PALBO and RIBO, particular attention was paid during method

development to this phenomenon. Carryover was evaluated as the peak area percentage of a blank sample injected after the ULOQ respect to the peak area of the LLOQ for each analyte. Carryover should not exceed 20% of LLOQ.

**Intra- and inter-day precision and accuracy.** As required by FDA and EMA guidelines, the method precision and accuracy were determined during a single working day (intra-day, 6 replicates for each QC concentration) and during 5 different working days (inter-day, 3 replicates for each QC concentration). The measured concentrations had to be within 15% of the nominal value and this had to be verified for at least 2 out of 3 QCs at each concentration level and in each run.

**Stability.** Bench-top and long-term stability was assessed to ensure that sample preparation and sample analysis, as well as the storage conditions applied do not affect the quantification of the analytes of interest. Stability tests were conducted using QCs prepared in triplicate at each concentration (L, M, H): bench-top stability was investigated after 4 h at room temperature; stability of the deproteinized QCs was evaluated in autosampler set at 4 ˚C re-analysing the samples 24, 48, and 72 h after the first injection; freeze/thaw stability was assessed by analysing three freshly prepared aliquots of each QCs concentration, and then again after one and two freeze/thaw cycles. Long term stability of PALBO, RIBO and LETRO was investigated both in plasma, to assess patients' samples stability after storage at -80˚C, and in solvent (methanol) to assess working solutions stability after storage at -20˚C. Stability tests were considered verified if the testing samples did not exceed ±15% from the nominal concentrations at each QCs concentration.

**Application of the method to clinical samples.** The proposed method was applied to quantify the $C_{min}$ of PALBO, RIBO and LETRO in samples from patients recruited from June to August 2019 in a clinical study (prot. code: CRO-2018-83) ongoing at the National Cancer Institute of Aviano, Italy. The inclusion criteria were: 1) to be under treatment with palbociclib or ribociclib according to the routine clinical practice criteria (the dose and the treatment cycle were not considered but patients should have been at the steady state); 2) age ≥18; 3) life expectancy > 3 months; 4) Signed informed consent. The exclusion criteria were to be non-collaborative and/or unreliable patients and refusal of informed consent. Patients were asked to periodically (every two months) collect plasma samples for the estimation of PALBO or RIBO and/or LETRO $C_{min}$ during their therapy treatment.

To accurately estimate the $C_{min}$ of PALBO, RIBO and LETRO, blood samples were collected immediately before the scheduled drug intake in 2.7 mL K-EDTA tubes. Plasma was obtained immediately by centrifugation of the blood samples at 3000 g for 10 min at 4˚C. Then the obtained plasma was split into two independent aliquots and stored at -80˚C in two different freezers.

**Ethics statement regarding patients' samples.** The clinical study (prot. code: CRO-2018-83) was approved by the local ethics committee (Comitato Etico Unico Regionale- C.E.U.R.) and it is conducted at the National Cancer Institute of Aviano (Italy). The study conduction fulfilled Declaration of Helsinki's principles. Patients were informed by the oncologist about the clinical study during their visits and were recruited only after the signature of a written informed consent. Patients' recruitment and data management and analysis are entirely conducted at the National Cancer Institute of Aviano (Italy).

**Reproducibility or incurred samples reanalysis.** At present, each of the 10 patients' samples collected were quantified in 2 separate runs during 2 different working days, to further assess the reproducibility of the proposed method. In fact, as recently introduced in the last version of the FDA guideline [23], incurred samples re-analysis (ISR) is necessary to show the reliability of the reported analyte concentrations obtained with a bioanalytical method. ISR is conducted by repeating the analysis of a subset of patients' samples in separate runs. The two

analyses can be considered equivalent if the percentage difference [expressed as: (repeat-original)*100/mean] between the first and the second concentration measured is within ±20% for at least 67% of the samples [23,24].

## Results and discussion

### LC-MS/MS method

Source dependent parameters were optimized as follows: temperature 500 ˚C, nebulizer gas 40 psi and heater gas 40 psi (zero air), curtain gas 35 psi and collision gas (CAD) 6 psi (nitrogen), ion spray voltage 5500 V. With the ESI source operating in positive ion mode, PALBO, RIBO, and LETRO formed in prevalence the protonated molecule [M+H]$^+$: 448 *m/z* for PALBO, 435 *m/z* for RIBO and 286 *m/z* for LETRO. The fragmentation patterns obtained within the collision cell are represented in Fig 1 [14,18,28] and reported in Table 1 along with the optimized compound dependent parameters. The daughter ions used as quantifiers were: 448>380 m/z for PALBO, 435>322 *m/z* for RIBO and 286>217 *m/z* for LETRO. The following fragment ions were used as qualifiers: 448>337 m/z for PALBO (in Fig 1a, the *m/z* 362 peak is probably due to the loss of $H_2O$), 435>367 *m/z* for RIBO and 286>190 *m/z* for LETRO. The quantifier/qualifier ions ratio was calculated across the calibration range (from LLOQ to A using 6 calibration curves) for each analyte: it resulted 20.3±3.0 for PALBO, 0.11±0.01 for RIBO and 1.9 ±0.2 for LETRO. With regard to RIBO, the fragment having lower intensity was chosen as quantifier ion to prevent signal saturation problem at higher concentrations.

The quantification of the ISs signal was conducted using the following transitions: 456>388 *m/z* for D$_8$-PALBO, 441>373 *m/z* for D$_6$-RIBO, and 290>221 *m/z* for $^{13}C_2,^{15}N_2$-LETRO.

The separation of the analytes was obtained applying the following gradient (flow rate of 0.3 mL/min, column temperature fixed at 50 ˚C): the percentage of MPB was increased from the initial condition (10%) to 70% in 0.5 min and then kept constant for 1.75 min; MPB was further increased to 95% in 0.1 min and kept constant for 1.9 min; the initial condition was then restored in 0.25 min and the column was re-equilibrated for 2 min. The total run time was 6.5 min. Fig 2 displays typical SRM chromatograms of plasma samples: an extracted blank sample (Fig 2a); an extracted sample at the LLOQ level (Fig 2b); an example of sample from a patient treated with PALBO (Fig 2c) and a sample from a patient treated with RIBO (600 mg/day) and LETRO (Fig 2d). The first sample was collected from a patient treated with PALBO at 125 mg/day after 25 h from the last capsule intake, while the second sample was collected from a patient treated with RIBO (600 mg/day) in combination with LETRO (2.5 mg/day) after 30 h from the last pills intake. As noticeable from Fig 2, the analytes were rapidly and selectively eluted achieving a good separation within 2.5 min: the retention times correspond to 1.93 min for PALBO, 1.56 min for RIBO and 2.20 min for LETRO. With an unretained peak time of 0.43 min, the retention factor (k) was 3.5 for PALBO, 2.6 for RIBO and 4.1 for LETRO.

### Recovery, matrix effect and selectivity

PALBO, RIBO and LETRO recovery, expressed as percentage and reported in Table 2 resulted high (≥ 92.3%) for all the analytes and reproducible over the concentrations ranges tested.

Both the post-column infusion test and the calculation of the ratio between analytes peak area in presence of matrix (human pooled plasma) and the peak area in absence of matrix (methanol) using QCs demonstrated the absence of significant matrix effect. In fact, no suppression or enhancement of extracted ions signals (XIC) was detected at the retention time of the analytes. The estimated matrix effect (ME%) is reported in Table 3 for each analyte: it was found to be between 91.5–98.7% with a CV% ≤10.2% for PALBO, between 85.0–113.2% with

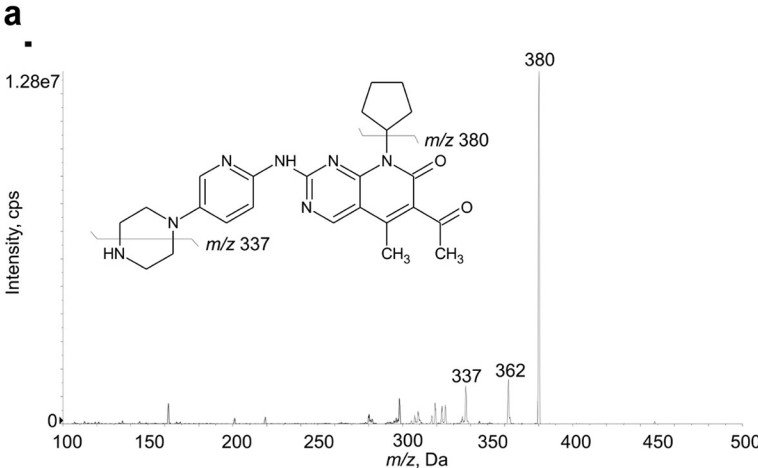

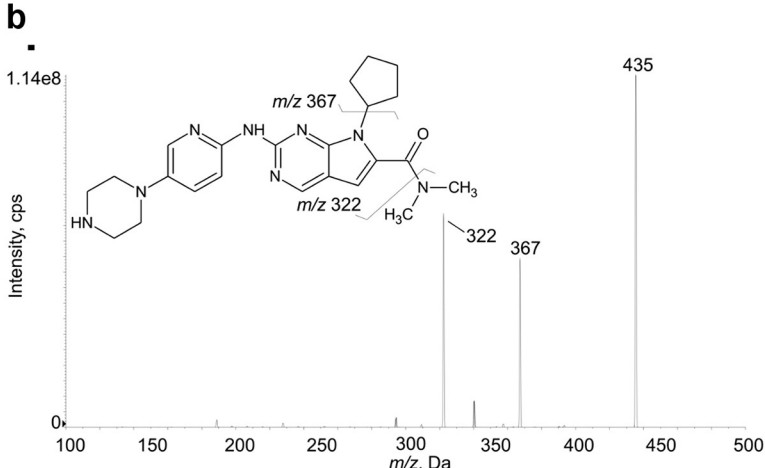

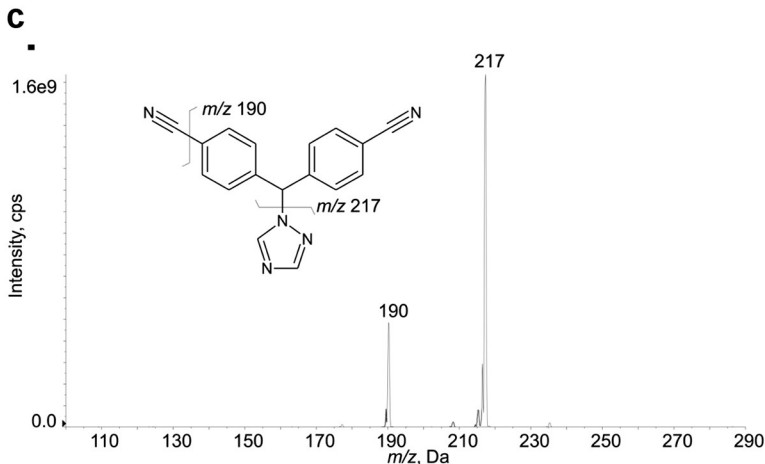

**Fig 1. MS/MS mass spectra of analytes with their chemical structures and identification of the fragment ions used for the present method.** (a) PALBO, recorded with CE = 50 V; (b) RIBO, recorded with CE = 37 V; (c) LETRO, recorded with CE = 30 V.

**Table 1. Compound-dependent parameters.**

| Compound | Q1[a] (m/z) | DP[b] (V) | EP[c] (V) | Q3[d] (m/z) | CE[e] (V) | CXP[f] (V) |
|---|---|---|---|---|---|---|
| **PALBO** | 448.3 | 130 | 10 | 380.2 | 40 | 10 |
| | | | | 337.2 | 53 | 10 |
| **RIBO** | 435.3 | 110 | 10 | 322.3 | 63 | 10 |
| | | | | 367.3 | 55 | 10 |
| **LETRO** | 286.2 | 50 | 10 | 217.2 | 20 | 10 |
| | | | | 190.2 | 45 | 10 |
| **D₈-PALBO** | 456.3 | 130 | 10 | 388.3 | 40 | 10 |
| **D₆-RIBO** | 441.3 | 110 | 10 | 373.3 | 38 | 10 |
| **¹³C₂,¹⁵N₂-LETRO** | 290.2 | 50 | 10 | 221.2 | 20 | 10 |

[a]First quadrupole mass.

[b]Declustering potential.

[c]Entrance potential.

[d]Third quadrupole mass.

[e]Collision energy.

[f]Cell exit potential.

a CV% ≤4.0% for RIBO and between 86.4–91.6% with a CV% ≤9.5 for LETRO. These results confirmed that the proposed method is not affected by matrix effect.

## Linearity and sensitivity

The linearity of the method was verified over the selected concentrations (0.3–250, 10–10000, 0.5–500 ng/mL for PALBO, RIBO and LETRO, respectively): the mean $R^2$ values obtained were 0.9990±0.0007 for PALBO, 0.9992±0.0002 for RIBO, and 0.9983±0.0010 for LETRO. A calibration curve example for each analyte is reported in Fig 3. As related to PALBO, the calculated accuracy was between 95.5 and 103.3% and precision was within 5.7%. The accuracy obtained for RIBO was between 95.1 and 102.7% while precision was ≤ 5.1%. Lastly, precision and accuracy of LETRO was between 91.8% and 104.5% and within 6.2%, respectively. In Table 4 the complete list of linearity data is reported.

The LLOQ values were assessed at the concentrations of 0.3 ng/mL for PALBO, 10 ng/mL for RIBO and 0.5 ng/mL for LETRO: the accuracy and precision (CV%) obtained for the 6 LLOQ samples prepared in pooled blank human plasma were, respectively, 98.1% and 6.5% for PALBO, 105.3% and 5.5% for RIBO and 108.2% and 4.4% for LETRO. The S/N ratios were 30.5 for PALBO, 93.5 for RIBO and 7.5 for LETRO, as reported in Fig 2 showing the chromatogram of a LLOQ sample with the corresponding S/N ratio for each analyte.

## Carryover

A marked carryover was observed after the injection of ULOQ sample, albeit the introduction of a cleaning step during the chromatographic gradient: the first blank sample after ULOQ had residual signals of PALBO and RIBO that were 2- and 1.5-fold higher than those of the LLOQ, respectively. On the contrary, no carryover post-injection was detected for LETRO, being its signal in the first blank sample after ULOQ injection lower than 10% respect to the LLOQ. The first attempt to overcome PALBO and RIBO carryover was the injection of several blank samples after the ULOQ. Unfortunately, after 7 blank samples carryover was still present (35% for PALBO and 30% for RIBO). Thus, keeping constant the mass spectrometry conditions and the MPA-B composition, we developed a washing method based on the "saw-tooth

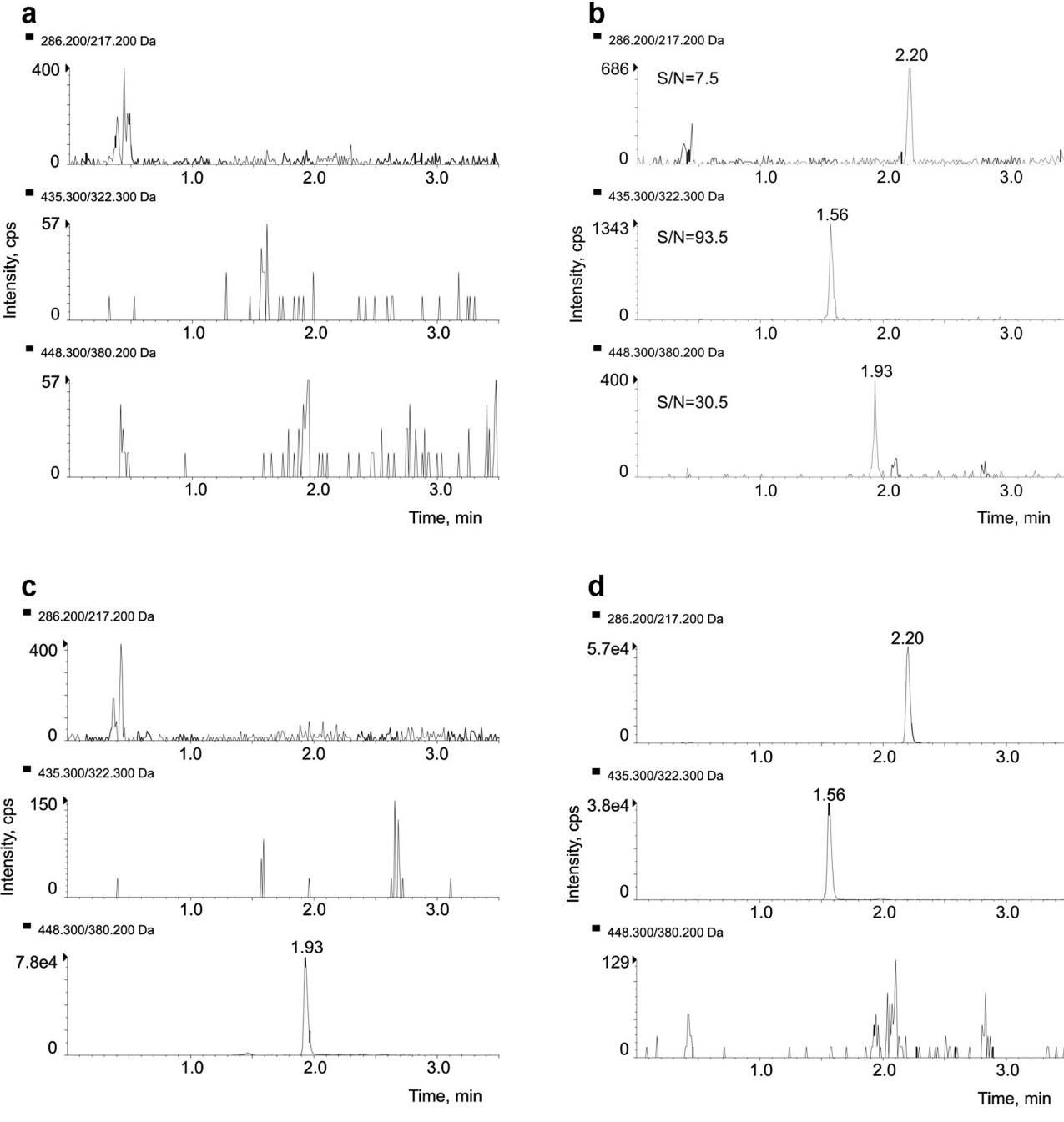

**Fig 2. SRM chromatograms.** (a) blank plasma sample; (b) S/N ratio of PALBO, RIBO and LETRO at the LLOQ (0.3, 10 and 0.5 ng/mL for PALBO, RIBO and LETRO, respectively); (c) extracted plasma sample of a patient treated with PALBO showing the drug at the concentration of 77.5 ng/mL; (d) extracted plasma sample of a patient treated with RIBO in combination with LETRO showing the drugs at the concentration of 396.0 ng/mL and 46.8 ng/mL, respectively.

wash" gradient proposed by Williams et al. [29] to be applied to blank samples after ULOQ and unknown patients' samples: from 10% to 98% of MPB (methanol/isopropanol 9:1, v/v, with 0.1% HCOOH) in 0.5 min and kept constant for 0.7 min; then from 98% to 5% of MPB in 0.1 min and kept constant for 0.8 min. The same profile was repeated three times overall with a final reconditioning step of 2 min at 10% of MPB. The total run time was 6.34 min (Fig 4).

**Table 2. Recovery of PALBO, RIBO and LETRO from human plasma.**

| Analyte | Nominal conc. (ng/mL) | Recovery (%) ±SD | Recovery CV(%) |
|---|---|---|---|
| PALBO | 0.5 | 92.3±9.4 | 10.2 |
| | 20 | 97.4±3.3 | 3.4 |
| | 200 | 96.6±3.5 | 3.6 |
| RIBO | 20 | 101.1±2.6 | 2.6 |
| | 800 | 97.7±2.8 | 2.9 |
| | 8000 | 99.6±1.4 | 1.4 |
| LETRO | 1 | 98.1±4.0 | 4.1 |
| | 40 | 97.0±1.7 | 1.7 |
| | 400 | 99.2±3.1 | 3.1 |

As a result, after the 2 blank samples run with the washing method, no quantifiable peaks of PALBO and peaks of RIBO ≤15% respect to the LLOQ were observed in the following blank sample run with the quantification method.

## Intra- and inter-day precision and accuracy

The results of intra- and inter-day precision and accuracy of the proposed method complied with FDA and EMA requirements (Table 5). As related to intra-day precision and accuracy, the obtained values were, respectively, ≤ 3.6% and between 94.5–112.3% for all three analytes. At the same time, inter-day precision and accuracy were ≤ 7.3% and between 94.5–112.9%.

## Stability

PALBO, RIBO, and LETRO stability in plasma matrix was verified under the following conditions: 1) after 4 h at room temperature, being precision and accuracy, respectively, within 12.2% and between 88.2% and 103.5% for the three analytes; 2) after 2 months of storage at -80 ˚C, being precision and accuracy, respectively, within 5.0% and between 88.3% and 105.5% for the three analytes. The deproteinized QCs were stable in autosampler set at 4 ˚C for 72 h as proved by precision and accuracy values obtained (≤ 7.8% and between 93.9% and 111.1% for all the compounds). PALBO, RIBO, and LETRO resulted stable after 2 freeze/thaw cycles (taking together the 3 drugs, precision and accuracy values were ≤ 14.2% and between 102.3–111.9%, respectively). Long term stability in methanol was verified after 2 months of storage at -20˚C: for all 3 compounds, precision and accuracy were within 4.3 and between 97.6–112.1%, respectively. In S1, S2 and S3 Tables complete stability data are reported.

**Table 3. Estimated matrix effect (ME%) of PALBO, RIBO and LETRO in deproteinized human plasma.**

| Analyte | Nominal conc. (ng/mL) | ME (%) ±SD | ME CV(%) |
|---|---|---|---|
| PALBO | 0.5 | 91.5±9.3 | 10.2 |
| | 20 | 98.7±1.4 | 1.4 |
| | 200 | 97.0±2.5 | 2.6 |
| RIBO | 20 | 113.2±4.5 | 4.0 |
| | 800 | 110.2±2.4 | 2.2 |
| | 8000 | 85.0±1.2 | 1.4 |
| LETRO | 1 | 86.4±8.2 | 9.5 |
| | 40 | 91.6±2.0 | 2.1 |
| | 400 | 90.5±1.9 | 2.1 |

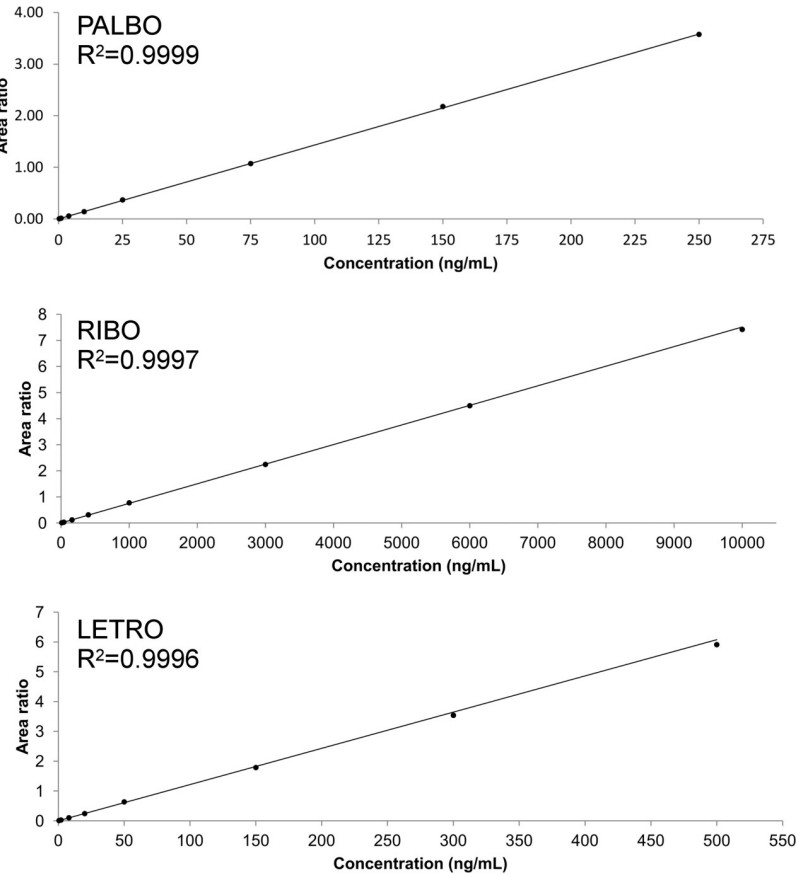

**Fig 3. Calibration curves of PALBO, RIBO and LETRO in human plasma.**

## Application of the method to clinical samples and reproducibility

Patients' recruitment into the clinical study (prot. code: CRO-2018-83) is still at the beginning and, at the moment, the method was tested on 10 plasma samples collected from 8 patients (from 2 patients we collected 2 sequential samples) affected by metastatic breast cancer. The principal demographic and clinical characteristics of the enrolled patients are reported in Table 6.

Blood samples were taken between 15 and 30 h from the last pill intake. In one case (patient 3, Table 7), the last PALBO assumption was 1 week before the blood sampling. The sensibility of the method allowed the quantification of the residual concentration of PALBO (1.6 ng/mL, I quantification). Concentrations of PALBO obtained in patients 1, 2, 3, 4, and 6 of Table 7 were rather in line with the population mean $C_{min}$ reported in literature (61 ng/mL) for the standard dose [4]. Patients 5 showed PALBO concentrations slightly lower in both two analyzed samples (39.5 and 41.5 ng/mL, I quantification, Table 7) while patient 7 showed a slightly higher drug concentration (97.9 ng/mL, I quantification, Table 7). This latter result was probably due to the fact that the last pill intake was 15 h before the sampling. In sample 8 (patient 6 of Table 7), although the patient was treated with PALBO in combination with fulvestrant, a residual amount of LETRO (2.8 ng/mL, I quantification) was detected: the patient completed her adjuvant therapy with LETRO 2 weeks before the sampling.

The concentrations of RIBO and LETRO obtained from the last sample (indicated in Table 7 as 5 and 6) resulted quite lower than the mean $C_{min}$ reported in literature for both

**Table 4. Accuracy and precision data of the calibration curves of PALBO, RIBO and LETRO.**

**PALBO (N[a] = 5)**
$R^2$ = 0.9990±0.0007
Intercept: -0.0009±0.0002
Slope: 0.0135±0.0001

| nominal conc. (ng/mL) | Mean ± SD | CV% | Acc% |
|---|---|---|---|
| 0.300 | 0.301±0.002 | 0.6 | 100.2 |
| 1.00 | 1.00±0.02 | 2.2 | 100.1 |
| 4.00 | 3.91±0.22 | 5.7 | 97.8 |
| 10.00 | 9.85±0.29 | 2.9 | 98.5 |
| 25.00 | 25.74±0.73 | 2.8 | 103.0 |
| 75.00 | 76.28±1.05 | 1.4 | 101.7 |
| 150.00 | 154.97±5.20 | 3.4 | 103.3 |
| 250.00 | 238.67±12.12 | 5.1 | 95.5 |

**RIBO (N[a] = 5)**
$R^2$ = 0.9992±0.0002
Intercept: 0.0008±0.0001
Slope: 0.00082±0.00004

| | | | |
|---|---|---|---|
| 10.00 | 10.04±0.13 | 1.3 | 100.4 |
| 40.00 | 39.30±2.00 | 5.1 | 98.2 |
| 160.00 | 158.44±3.68 | 2.3 | 99.0 |
| 400.00 | 410.94±11.23 | 2.7 | 102.7 |
| 1000.00 | 1020.18±19.12 | 1.9 | 102.0 |
| 3000.00 | 3029.34±88.65 | 2.9 | 101.0 |
| 6000.00 | 6091.68±176.43 | 2.9 | 101.5 |
| 10000.00 | 9506.24±299.76 | 3.2 | 95.1 |

**LETRO (N[a] = 5)**
$R^2$ = 0.9983±0.0010
Intercept: 0.0022±0.0006
Slope: 0.0122±0.0002

| | | | |
|---|---|---|---|
| 0.50 | 0.49±0.01 | 1.6 | 98.6 |
| 2.00 | 2.09±0.12 | 5.7 | 104.5 |
| 8.00 | 8.25±0.20 | 2.4 | 103.2 |
| 20.00 | 20.74±0.45 | 2.1 | 103.7 |
| 50.00 | 50.05±1.30 | 2.6 | 100.1 |
| 150.00 | 150.70±6.02 | 4.0 | 100.5 |
| 300.00 | 293.00±18.07 | 6.2 | 97.7 |
| 500.00 | 459.19±11.19 | 2.4 | 91.8 |

[a]Number of calibration curves used for accuracy and precision estimation.

drugs: RIBO concentration was 396.0 ng/mL while the population mean is 711 ng/mL [19], LETRO was measured at the concentration of 46.8 ng/mL while the reported mean is 107.0 ng/mL [11].

The quantifier/qualifier ions ratio calculated, for each analyte, in these patients' samples was in line with those obtained from calibration curves and QCs: 20.8±2.2 for PALBO (reference value 20.3±3.0), 0.11 for the single RIBO sample (reference value 0.11±0.01) and 2.0±0.2 for LETRO (reference value 1.9±0.2).

Each of these 10 patients samples collected and analysed at present were further quantified in a second run, to assess the reproducibility of the proposed method by means of the ISR. In Table 7 ISR data are reported: the concentrations of PALBO, RIBO and LETRO obtained with the first and the second quantification along with the percentage differences calculated.

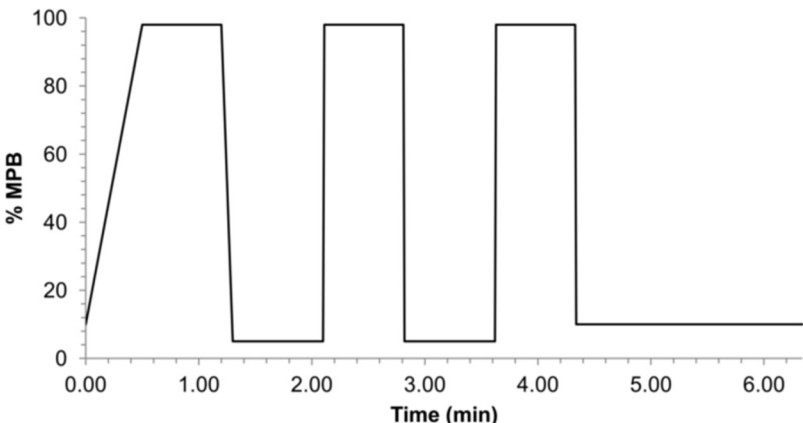

**Fig 4. Chromatographic gradient used for the washing method.** MPB is methanol/isopropanol 9:1, v/v, with 0.1% HCOOH; MPA is ultrapure water with 0.1% HCOOH.

**Table 5. Intra- and inter-day precision and accuracy of the proposed method for PALBO, RIBO and LETRO.**

Intra-day (N = 6)

| Analyte | Nominal conc. (ng/mL) | Mean ± SD | CV % | Acc % |
|---|---|---|---|---|
| PALBO | 0.50 | 0.56±0.02 | 4.0 | 112.3 |
| | 20.00 | 19.46±0.72 | 3.7 | 97.3 |
| | 200.00 | 195.03±7.51 | 3.9 | 97.5 |
| RIBO | 20.00 | 19.73±0.70 | 3.6 | 98.7 |
| | 800.00 | 757.56±37.94 | 5.0 | 94.7 |
| | 8000.00 | 7560.69±334.39 | 4.4 | 94.5 |
| LETRO | 1.00 | 1.00±0.06 | 6.3 | 100.2 |
| | 40.00 | 38.37±1.24 | 3.2 | 95.9 |
| | 400.00 | 382.52±13.21 | 3.5 | 95.6 |

Inter-day (N = 15)

| Analyte | Nominal conc. (ng/mL) | Mean ± SD | CV % | Acc % |
|---|---|---|---|---|
| PALBO | 0.50 | 0.56±0.04 | 6.2 | 112.9 |
| | 20.00 | 21.12±1.13 | 5.4 | 105.6 |
| | 200.00 | 206.72±8.66 | 4.2 | 103.4 |
| RIBO | 20.00 | 20.23±1.17 | 5.8 | 101.1 |
| | 800.00 | 794.70±42.33 | 5.3 | 99.3 |
| | 8000.00 | 7757.54±354.15 | 4.6 | 97.0 |
| LETRO | 1.00 | 0.95±0.07 | 7.3 | 94.5 |
| | 40.00 | 41.10±1.79 | 4.4 | 102.8 |
| | 400.00 | 394.74±20.09 | 5.1 | 98.7 |

**Table 6. Principal demographic and clinical characteristics of the enrolled patients.**

| Patients characteristic | N |
|---|---|
| Population size and sex | 8 female |
| Age (range) | 67 (50–85) years |
| Therapy | 5 PALBO (125 mg/day) + fulvestrant<br>1 PALBO (100 mg/day) + fulvestrant<br>1 PALBO (125 mg/day) + LETRO (2.5 mg/day)<br>1 RIBO (600 mg/day) + LETRO (2.5 mg/day) |

**Table 7. Incurred samples reanalysis.**

| Patient N | Sample N | PALBO (mg/day) | RIBO (mg/day) | LETRO (mg/day) | I quantif.[a] (ng/mL) | II quantif.[a] (ng/mL) | % diff.[b] |
|---|---|---|---|---|---|---|---|
| 1 | 1 | 125 | - | - | 53.0 | 55.9 | 5.3 |
| 2 | 2 | 125 | - | - | 76.9 | 84.2 | 9.1 |
| 2 | 3 | 125 | - | - | 77.5 | 82.4 | 6.1 |
| 3 | 4 | 125 | - | - | 1.6 | 1.5 | -6.5 |
| 4 | 5 | 100 | - | - | 70.7 | 74.3 | 5.0 |
| 5 | 6 | 125 | - | - | 39.5 | 43.0 | 8.5 |
| 5 | 7 | 125 | - | - | 41.5 | 46.1 | 10.5 |
| 6 | 8 | 125 | - | - | 69.7 | 74.0 | 6.0 |
| 6 | 8 | - | - | 2.5 | 63.3 | 67.8 | 6.9 |
| 7 | 9 | 125 | - | - | 97.9 | 96.2 | -1.8 |
| 7 | 9 | - | - | 2.5 | 2.8 | 2.9 | 3.5 |
| 8 | 10 | - | 600 | - | 396.0 | 419.3 | 5.7 |
| 8 | 10 | - | - | 2.5 | 46.8 | 46.9 | 0.2 |

[a]Drugs' concentrations (ng/mL) quantified during the I and II analysis. (

[b]Percentage difference between the I and II quantification.

Despite this test is limited by the low number of available samples, preliminary results seemed to indicate a good reproducibility of the method: the percentage differences were always within ±10% for all the analytes (between -6.5% and 10.5%) and the $R^2$ of the correlation graph between the two quantifications was 0.9994 (Fig 5).

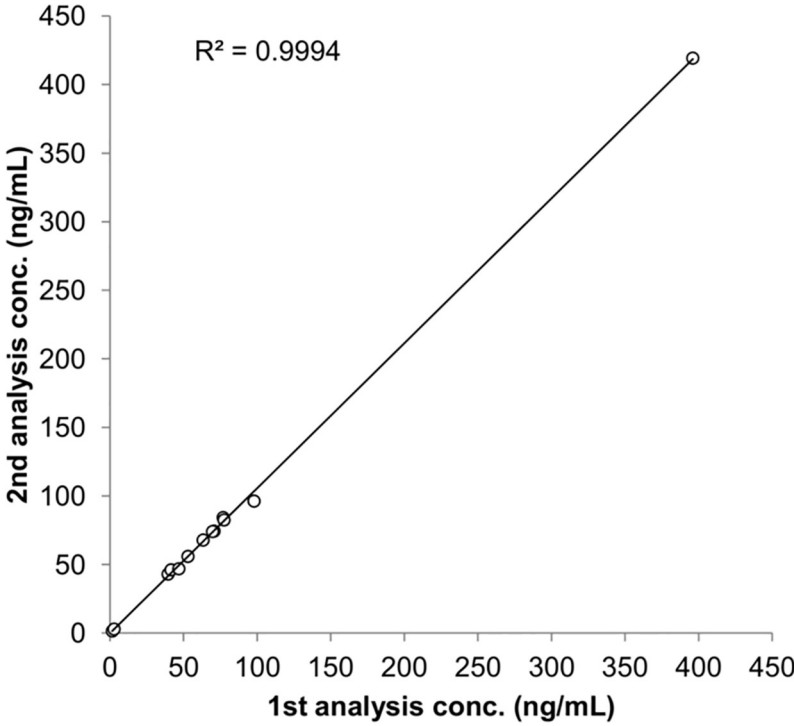

**Fig 5. Correlation graph between the first and the second analysis of PALBO, RIBO and LETRO in patients' samples (N = 6).**

## Conclusion

The first LC-MS/MS method for the simultaneous quantification of PALBO, RIBO and LETRO in human plasma was developed and successfully validated according to FDA/EMA guidelines [23,24]. Once overcome the PALBO and RIBO carryover issue by introducing blank samples to be run with a specifically developed washing method, calibration curves properly covered the *in vivo* concentrations of the drugs [6,11,19–22]. The proposed method resulted linear over the concentration ranges of 0.3–250 ng/mL for PALBO, 10–10000 ng/mL for RIBO and 0.5–500 ng/mL for LETRO, while the only previously published method for the quantification of these CDKIs narrowed the concentration range to 2–200 ng/mL due to the carryover problem [18]. This method was applied to quantify $C_{min}$ of PALBO, RIBO and LETRO in 10 plasma samples from patients enrolled in a clinical study (CRO-2018-83) ongoing at the National Cancer Institute of Aviano. The ample calibration curve ranges allow to apply the proposed method in order to evaluate also other pharmacokinetic parameters or clinical query: among the study samples analysed, PALBO was detected at the concentration of 1.6 ng/mL after 1 week off treatment and LETRO was detected (2.8 ng/mL) after 2 week from the last pill intake. This simple and versatile analytical method could be an useful instrument to promote the personalization of the anticancer therapy.

## Supporting information

**S1 Table. Short term stability of PALBO, RIBO and LETRO.**
(DOCX)

**S2 Table. Stability after two freeze-thaw cycles.**
(DOCX)

**S3 Table. Long term stability (2 months) of PALBO, RIBO and LETRO: Analytes stored in human plasma at -80˚C and working solutions (methanol) stored at -20˚C.**
(DOCX)

## Acknowledgments

We warmly thank the patients for their participation in the clinical study and Dr. Sara Colò for her valuable assistance in revising English language. We also thank the "Ministero della Salute Ricerca Corrente" for its support.

## Author Contributions

**Conceptualization:** Fabio Puglisi, Elena Marangon, Giuseppe Toffoli.

**Investigation:** Bianca Posocco, Mauro Buzzo, Ariana Soledad Poetto, Marco Orleni, Sara Gagno, Martina Zanchetta, Valentina Iacuzzi.

**Methodology:** Bianca Posocco, Mauro Buzzo, Ariana Soledad Poetto, Elena Marangon.

**Resources:** Michela Guardascione, Fabio Puglisi, Debora Basile, Giacomo Pelizzari.

**Supervision:** Giuseppe Toffoli.

**Writing – original draft:** Bianca Posocco.

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
