## [Decision Letter · Decision Letter 0]

6 Nov 2019

PONE-D-19-26156

Simultaneous quantification of palbociclib, ribociclib and letrozole in human plasma by a new LC-MS/MS method for clinical application

PLOS ONE

Dear Dr. Posocco,

Thank you for submitting your manuscript to PLOS ONE. After careful consideration, we feel that it has merit but does not fully meet PLOS ONE’s publication criteria as it currently stands. Therefore, we invite you to submit a revised version of the manuscript that addresses the points raised during the review process.

Please respond to all of the reviewer queries and provide all of the details for the method and the results that they have requested.  I have assigned this manuscript as minor revision, because no additional experiments appear to be necessary, but you will need to fully address the concerns raised by the reviewers.  As noted by one reviewer, it is helpful to the readers to have the manuscript evaluated by a colleague or editorial service for English grammar and usage, particularly as PLoS ONE does not edit manuscripts in this way prior to publication.

We would appreciate receiving your revised manuscript by Dec 21 2019 11:59PM. To enhance the reproducibility of your results, we recommend that if applicable you deposit your laboratory protocols in protocols.io, where a protocol can be assigned its own identifier (DOI) such that it can be cited independently in the future. For instructions see: http://journals.plos.org/plosone/s/submission-guidelines#loc-laboratory-protocols

We look forward to receiving your revised manuscript.

Kind regards,

John Matthew Koomen, PhD

Academic Editor

PLOS ONE

Journal Requirements:

2. Please provide the product and lot numbers (if provided) of the palbociclib provided by Toronto Research Chemicals and Ribociclib and Letrozole provided by Merck-Sigma Aldrich.

3. In your Methods section, please provide additional information about the participant recruitment method and the demographic details of your participants. Please ensure you have provided sufficient details to replicate the analyses such as: a) a description of how participants were recruited, and b) descriptions of where participants were recruited and where the research took place.

I have read the journal's policy and the authors of this manuscript have the following competing interests: Dr. Fabio Puglisi reports grants from Astrazeneca and from Roche,  outside the submitted work.

Reviewers' comments:

Reviewer's Responses to Questions

**Comments to the Author**

1. Is the manuscript technically sound, and do the data support the conclusions?

Reviewer #1: Partly

Reviewer #2: Yes

2. Has the statistical analysis been performed appropriately and rigorously? 

Reviewer #1: Yes

Reviewer #2: Yes

3. Have the authors made all data underlying the findings in their manuscript fully available?

Reviewer #1: Yes

Reviewer #2: Yes

4. Is the manuscript presented in an intelligible fashion and written in standard English?

Reviewer #1: Yes

Reviewer #2: Yes

5. Review Comments to the Author

Reviewer #1: The authors present a quantitative LC/MS/MS method for the analysis of cyclin dependent 22 kinase inhibitors (CDKIs) palbociclib and ribociclib and the aromatase inhibitor letrozole in human plasma. The indicated assay has potential clinical utility with a short run time of 6.5 minutes and an analytical range that may support therapeutic drug monitoring for cancer patients receiving these drugs. The authors in future might consider reducing the calibration range to avoid carryover problems and then dilute anticipated higher concentration patient specimens into this range. Reducing the calibration range may lead to the use of linear regression vs the indicated quadratic regression. Parameters such as recovery, precision, accuracy, incurred sample reanalysis were adequately presented by the authors as per FDA and EMA guidance. Points of concern are described below.

Major issues:

1) Page 7 row 161. Please include a figure of the calibration curves for all analytes and a statement about why you are using quadratic curve fitting, and not linear. MS assays should ideally be linear. Small changes in signal intensity with a quadratic calibration curve can potentially lead to big changes in the calculated concentration in comparison to linear regression fitting.

2) Page 14 line 301. Was the source of carryover determined and could this be overcome with a needle washing or valve washing step during analysis? At what concentration was carryover not observed for PALBO and RIBO? Implementation for routine clinical use of this assay would require that carryover be effectively removed in the original sample injection.

3) Matrix effects were evaluated with healthy donor plasma. Were matrix effects observed if experiments were performed with patient specimen, lipemic plasma, or hemolyzed plasma?

4) Qualifier and quantifier MRMs were indicated. Please present data on the consistency of the quantifier/qualifier ratios across the indicated calibration range and also in patient specimens within batch(s).

5) The presented patient sample sized is quite small. Were there specimens from the same patients (or different patients) taking the indicated drugs collected near Cmax and analyzed using the presented calibration range? Or dilution of specimens near Cmax into the calibration range. Patient specimen 8 was above the indicated PALBO calibration range, was this sample diluted into the calibration range?

Minor issues and comments:

1) Figure quality is low, please input as higher dpi images

2) Table 6 second row should be “Population size and sex”

3) Page 4 row 75. Remove the word “actually”

4) Page 4 row 78. Change “185,5” to “185.5”

5) Page 7 row 152. Replace “Then, matrix effect was” with “The matrix effect was then”

6) Page 7 row 153 “The CV should be within 15%.” Include a reference or explanation as to why this should be 15%.

7) Page 7 row 166-167. Insert references (eg FDA and EMA guidance documents) regarding the pre-determined acceptance criteria

8) Page 7 row 171. Please indicate the method of S/N calculation

9) Page 9 row 219. Remove “till now”.

10) Page 10 row 226. Include ref to FDA and EMA regarding this criteria.

11) Page 11 row 249. Change “mg/die” to “mg/day” throughout paper.

12) Page 11 line 252. Calculate the retention factor (k) and include values for analytes.

13) Page 13 Table 4. Indicate if the N is from inter- or intra-day analysis of the calibration curve. Please also indicate the (Avg±SD) equation of the calibration curve regression model used across multiple days of analysis.

14) Page 16 line 348. Remove word “Anyway”

15) Please check entire document for grammatical errors.

Reviewer #2: Thank you for your submission. This manuscript is well written and statistically sound.

While you have increased the concentration range of your compounds, this use of the carryover wash step makes the method not feasibility well constructed for clinical application in certain environments. If I am understanding the method correctly, (line 314) two additional wash step runs need to be included after each unknown sample? This can cause extensive waste of mobile phases, column wear over time as well as instrument issues as the total run time of the entire batch will triple.

Line 310 says the washing method was developed to be utilized after ULOQ and unknown patient samples, however Line 314 says two of these washing methods are run back to back to prove no carryover exists. Please make clear whether one or two blank samples run with the washing method are essential after each unknown and ULOQ, for reproducibility purposes.

6. PLOS authors have the option to publish the peer review history of their article (what does this mean?). If published, this will include your full peer review and any attached files.

Reviewer #1: No

Reviewer #2: No

---

## [Author Response · Author response to Decision Letter 0]

20 Dec 2019

Review Comments to the Author

Reviewer #1: The authors present a quantitative LC/MS/MS method for the analysis of cyclin dependent 22 kinase inhibitors (CDKIs) palbociclib and ribociclib and the aromatase inhibitor letrozole in human plasma. The indicated assay has potential clinical utility with a short run time of 6.5 minutes and an analytical range that may support therapeutic drug monitoring for cancer patients receiving these drugs. The authors in future might consider reducing the calibration range to avoid carryover problems and then dilute anticipated higher concentration patient specimens into this range. Reducing the calibration range may lead to the use of linear regression vs the indicated quadratic regression. Parameters such as recovery, precision, accuracy, incurred sample reanalysis were adequately presented by the authors as per FDA and EMA guidance. Points of concern are described below.

Major issues:

1) Page 7 row 161. Please include a figure of the calibration curves for all analytes and a statement about why you are using quadratic curve fitting, and not linear. MS assays should ideally be linear. Small changes in signal intensity with a quadratic calibration curve can potentially lead to big changes in the calculated concentration in comparison to linear regression fitting.

As correctly pointed out by the Rewiever, the best curve fitting for MS assay should be the linear one. The use of a quadratic regression has been erroneously reported within the manuscript. Instead, a linear regression model has been used applying a weighted factor of 1/x2, due to the heteroscedatic error of the instrument (the calibration curve equation of each analyte has been reported). We apologize for the scarce clarity in this paragraph. An additional figure reporting example of calibration curve for each analytes has been added to the manuscript (Fig 3).

2) Page 14 line 301. Was the source of carryover determined and could this be overcome with a needle washing or valve washing step during analysis? At what concentration was carryover not observed for PALBO and RIBO? Implementation for routine clinical use of this assay would require that carryover be effectively removed in the original sample injection.

As properly underlined by the Reviewer, carryover should be completely removed in order to apply the proposed method to routine clinical use. To address the palbociclib and ribociclib carryover effect several attempts have been made, including: 1) increasing of rinse DIP time; 2) increasing of the rinse volume up to 2 mL with a loop of 50 µL; 3) testing different needle washing solutions. Nonetheless, the carryover effect was still present probably because its source was the column. The possibility to apply a different column type was also investigated. Unfortunately, the carryover effect was still present and the best performance, in terms of run time and peak separation, was obtained with the Luna Omega Polar C18 column. The concentration range reduction was not an option, in this case, since to reduce (<20% respect to the LLOQ) palbociclib and ribociclib signals in the blank sample run after the ULOQ we should not exceed 25 ng/mL for palbociclib and 1500 ng/mL for ribociclib. For this reason we added two washing blank samples after the ULOQ and after each unknown sample. As a result, the following blank sample was free from carryover effect.

3) Matrix effects were evaluated with healthy donor plasma. Were matrix effects observed if experiments were performed with patient specimen, lipemic plasma, or hemolyzed plasma?

We do test the matrix effect in haemolysed plasma, since it is not infrequent, in our experience, that both patients’ samples and healthy donors’ plasma are haemolysed. No matrix effect was observed in haemolysed plasma and pooled plasma (18 different healthy donor) used for calibration curves and QCs samples is composed also of some haemolysed plasma samples.

On the other hand, matrix effect was not tested in lipemic plasma or patients’ samples. This choice was determined by two reasons: 1) according to the clinical protocol blood sampling was conducted in association with routine blood tests and patients were asked to be fasting (thus reducing the incidence of lipemic plasma); 2) plasma matrix can be quite variable in lipid content during a whole PK curve while we collected only one specimen per patient with a specific timing and condition. Considering the overall low frequency of lipemic samples (0.5-2.5% from our experience and literature data) in addition to the peculiar blood sampling conditions, we considered the matrix effect test on lipemic plasma not needed (although required by the guidelines and properly remarked by the Reviewer). 

It was not possible to test matrix effect in patients’ specimens since the clinical protocol was designed to collect blood samples containing drugs at the Cmin value (no free-drug pre-dose sampling).

4) Qualifier and quantifier MRMs were indicated. Please present data on the consistency of the quantifier/qualifier ratios across the indicated calibration range and also in patient specimens within batch(s).

The quantifier/qualifier ions ratio was calculated for each analyte and reported within the Results section (“LC-MS/MS method” paragraph, page 10) with the following sentence: “The quantifier/qualifier ions ratio was calculated across the calibration range (from LLOQ to A using 6 calibration curves) for each analyte: it resulted 20.3±3.0 for PALBO, 0.11±0.01 for RIBO and 1.9±0.2 for LETRO. Regarding RIBO, the fragment having lower intensity was chosen as quantifier ion to prevent signal saturation problem at higher concentrations.”

These values were confirmed in patients’ samples, as reported in Results section (“Application of the method to clinical samples and reproducibility” paragraph, page 18): “The quantifier/qualifier ions ratio calculated, for each analyte, in these patients’ samples was in line with those obtained from calibration curves and QCs: 20.8±2.2 for PALBO (reference value 20.3±3.0), 0.11 for the single RIBO sample (reference value 0.11±0.01) and 2.0±0.2 for LETRO (reference value 1.9±0.2).”

5) The presented patient sample sized is quite small. Were there specimens from the same patients (or different patients) taking the indicated drugs collected near Cmax and analyzed using the presented calibration range? Or dilution of specimens near Cmax into the calibration range. Patient specimen 8 was above the indicated PALBO calibration range, was this sample diluted into the calibration range?

The Reviewer’s comment regarding the small samples size is proper. Nonetheless, patients were recruited over only three months and the IRCCS C.R.O of Aviano was the only enrolling centre according to the clinical protocol. Blood samples were taken as near as possible to the Cmin (i.e. 24 h after the last pill intake). In fact all specimens (from both the same patient or different patients) were taken between 15 and 30 h from the last pill intake. We thank the Reviewer for his/her question regarding patient 8: there is actually a mistake in Table 7 since this patient was treated with ribociclib 600 mg/day (instead of palbociclib), thus the concentrations reported in Table (396.0 and 419.3 ng/mL) are related to this drug and are within the calibration range (10-10000 ng/mL). We apologize for the mistake. 

Minor issues and comments:

1) Figure quality is low, please input as higher dpi images

Figure has been revised and checked with PACE.

2) Table 6 second row should be “Population size and sex”

Table 6 has been corrected as indicated by Reviewer #1.

3) Page 4 row 75. Remove the word “actually”

Done. 

4) Page 4 row 78. Change “185,5” to “185.5”

Done.

5) Page 7 row 152. Replace “Then, matrix effect was” with “The matrix effect was then”

Done.

6) Page 7 row 153 “The CV should be within 15%.” Include a reference or explanation as to why this should be 15%.

The appropriate reference has been added to the text.

7) Page 7 row 166-167. Insert references (eg FDA and EMA guidance documents) regarding the pre-determined acceptance criteria

Done.

8) Page 7 row 171. Please indicate the method of S/N calculation

The Analyst software calculates the S/N ratio using Peak-to-Peak method taking the standard deviation of all the chromatographic data points between the specified background start and background end times (60 min before analyte peak). This sentence has been added to the text. 

9) Page 9 row 219. Remove “till now”.

Done.

10) Page 10 row 226. Include ref to FDA and EMA regarding this criteria.

The appropriate references has been added to the text.

11) Page 11 row 249. Change “mg/die” to “mg/day” throughout paper.

Done.

12) Page 11 line 252. Calculate the retention factor (k) and include values for analytes.

Retention factors has been calculated and reported as indicated by the Reviewer. The following sentence has been added to the text: “With an unretained peak time of 0.43 min, the retention factor (k) was 3.5 for PALBO, 2.6 for RIBO and 4.1 for LETRO.”

13) Page 13 Table 4. Indicate if the N is from inter- or intra-day analysis of the calibration curve. Please also indicate the (Avg±SD) equation of the calibration curve regression model used across multiple days of analysis.

The 5 calibration curves used for the linearity estimation and reported in Table 4 with “N=5” belong both to the inter- and intra-day assessments. In fact, we used the first 4 inter-day assessments and the intra-day assessment. For this reason we did not specify whether N was from inter- or intra-day. Moreover, for clarity, we preferred not to introduce the concept of inter- and intra-day assessment before the corresponding paragraph. We hope this answer could satisfy the Reviewer’s requirement.

The calibration curve equation (Avg±SD) has been reported for each analyte within Table 4.

14) Page 16 line 348. Remove word “Anyway”

Done.

15) Please check entire document for grammatical errors.

The manuscript has been revised for English grammar.

Reviewer #2: Thank you for your submission. This manuscript is well written and statistically sound.

While you have increased the concentration range of your compounds, this use of the carryover wash step makes the method not feasibility well constructed for clinical application in certain environments. If I am understanding the method correctly, (line 314) two additional wash step runs need to be included after each unknown sample? This can cause extensive waste of mobile phases, column wear over time as well as instrument issues as the total run time of the entire batch will triple.

As properly underlined by the Reviewer, carryover should be completely removed in order to apply the proposed method to routine clinical use. Several attempts have been made to address the palbociclib and ribociclib carryover effect, including: 1) increasing of rinse DIP time; 2) increasing of the rinse volume up to 2 mL with a loop of 50 µL; 3) testing different needle washing solutions. Nonetheless, the carryover effect was still present probably because its source was the column. The possibility to apply a different column type was also investigated. Unfortunately, the carryover effect was still present and the best performance, in terms of run time and peak separation, was obtained with the Luna Omega Polar C18 column. The concentration range reduction was not an option, in this case, since to reduce (<20% respect to the LLOQ) palbociclib and ribociclib signals in the blank sample run after the ULOQ we should not exceed 25 ng/mL for palbociclib and 1500 ng/mL for ribociclib. Such a calibration range reduction determines the dilution necessity of the most of patients’ samples. For this reason we added two washing blank samples after the ULOQ and after each unknown sample. As a result, the following blank sample was free from carryover effect. 

Line 310 says the washing method was developed to be utilized after ULOQ and unknown patient samples, however Line 314 says two of these washing methods are run back to back to prove no carryover exists. Please make clear whether one or two blank samples run with the washing method are essential after each unknown and ULOQ, for reproducibility purposes.

We apologize for the lack in clarity in the paragraph indicated by the Reviewer. The carryover was removed with 2 blank samples run with the specified washing method and a the following blank sample (a test sample for carryover) run with the quantification method showed no quantifiable peaks of PALBO and peaks of RIBO ≤15% respect to the LLOQ. Thus after ULOQ and each patient’s sample only two additional blank samples (washing method) needed to be run. Line 310 has been modified to increase the clarity for reproducible purpose, as suggested by the Reviewer.

---

## [Decision Letter · Decision Letter 1]

24 Jan 2020

Simultaneous quantification of palbociclib, ribociclib and letrozole in human plasma by a new LC-MS/MS method for clinical application

PONE-D-19-26156R1

Dear Dr. Posocco,

We are pleased to inform you that your manuscript has been judged scientifically suitable for publication and will be formally accepted for publication once it complies with all outstanding technical requirements.

With kind regards,

John Matthew Koomen, PhD

Academic Editor

PLOS ONE

Additional Editor Comments (optional):

Reviewers' comments:

Reviewer's Responses to Questions

**Comments to the Author**

1. If the authors have adequately addressed your comments raised in a previous round of review and you feel that this manuscript is now acceptable for publication, you may indicate that here to bypass the “Comments to the Author” section, enter your conflict of interest statement in the “Confidential to Editor” section, and submit your "Accept" recommendation.

Reviewer #1: All comments have been addressed

Reviewer #2: All comments have been addressed

2. Is the manuscript technically sound, and do the data support the conclusions?

Reviewer #1: Yes

Reviewer #2: (No Response)

3. Has the statistical analysis been performed appropriately and rigorously? 

Reviewer #1: Yes

Reviewer #2: (No Response)

4. Have the authors made all data underlying the findings in their manuscript fully available?

Reviewer #1: Yes

Reviewer #2: (No Response)

5. Is the manuscript presented in an intelligible fashion and written in standard English?

Reviewer #1: Yes

Reviewer #2: (No Response)

6. Review Comments to the Author

Reviewer #1: The authors have adequately addressed the concerns for this manuscript. Alternative approaches to reducing carryover for these analytes should be considered in future.

Reviewer #2: (No Response)

7. PLOS authors have the option to publish the peer review history of their article (what does this mean?). If published, this will include your full peer review and any attached files.

Reviewer #1: No

Reviewer #2: No

---

## [Editor Report · Acceptance letter]

30 Jan 2020

PONE-D-19-26156R1 

Simultaneous quantification of palbociclib, ribociclib and letrozole in human plasma by a new LC-MS/MS method for clinical application 

Dear Dr. Posocco:

I am pleased to inform you that your manuscript has been deemed suitable for publication in PLOS ONE. Congratulations! Your manuscript is now with our production department. 

With kind regards,

on behalf of

Dr. John Matthew Koomen 

Academic Editor

PLOS ONE